# MILE: A Multi-Level Framework for Scalable Graph Embedding

## Abstract

Recently there has been a surge of interest in designing graph embedding methods. Few, if any, can scale to a large-sized graph with millions of nodes due to both computational complexity and memory requirements. In this paper, we relax this limitation by introducing the MultI-Level Embedding (MILE) framework – a generic methodology allowing contemporary graph embedding methods to scale to large graphs. MILE repeatedly coarsens the graph into smaller ones using a hybrid matching technique to maintain the backbone structure of the graph. It then applies existing embedding methods on the coarsest graph and refines the embeddings to the original graph through a graph convolution neural network that it learns. The proposed MILE framework is agnostic to the underlying graph embedding techniques and can be applied to many existing graph embedding methods without modifying them. We employ our framework on several popular graph embedding techniques and conduct embedding for real-world graphs. Experimental results on five large-scale datasets demonstrate that MILE significantly boosts the speed (order of magnitude) of graph embedding while generating embeddings of better quality, for the task of node classification. MILE can comfortably scale to a graph with 9 million nodes and 40 million edges, on which existing methods run out of memory or take too long to compute on a modern workstation.

## 1 Introduction

In recent years, graph embedding has attracted much interest due to its broad applicability for various tasks (Perozzi et al., 2014; Wang et al., 2016; Henderson et al., 2012). However, such methods rarely scale to large datasets (e.g., graphs with over 1 million nodes) since they are computationally expensive and often memory intensive. For example, random-walk-based embedding techniques require a large amount of CPU time to generate a sufficient number of walks and train the embedding model. As another example, embedding methods based on matrix factorization, including GraRep (Cao et al., 2015) and NetMF (Qiu et al., 2018), requires constructing an enormous objective matrix (usually much denser than adjacency matrix), on which matrix factorization is performed. Even a medium-size graph with 100K nodes can easily require hundreds of GB of memory using those methods. On the other hand, many graph datasets in the real world tend to be large-scale with millions or even billions of nodes. To the best of our knowledge, none of the existing efforts examines how to scale up graph embedding in a **generic** way. We make the first attempt to close this gap. We are also interested in the related question of whether the quality of such embeddings can be improved along the way. Specifically, we ask:

1) Can we scale up the existing embedding techniques in an agnostic manner so that they can be directly applied to larger datasets?

2) Can the quality of such embedding methods be strengthened by incorporating the holistic view of the graph?

To tackle these problems, we propose a MultI-Level Embedding (MILE) framework for graph embedding. Our approach relies on a three-step process: **first**, we repeatedly coarsen the original graph into smaller ones by employing a hybrid matching strategy; **second**, we

compute the embeddings on the coarsest graph using an existing embedding techniques - and **third**, we propose a novel refinement model based on learning a graph convolution network to refine the embeddings from the coarsest graph to the original graph – learning a graph convolution network allows us to compute a refinement procedure that levers the dependencies inherent to the graph structure and the embedding method of choice. To summarize, we find that:

- MILE is generalizable : Our MILE framework is agnostic to the underlying graph embedding techniques and treats them as black boxes.

- MILE is scalable : MILE can *significantly improve the scalability of the embedding methods* (**up to 30-fold**), by reducing the running time and memory consumption.

- MILE generates high-quality embeddings : In many cases, we find that the quality of embeddings improves by levering MILE (in some cases is in excess of 10%).

## 2 Related Work

Many techniques for graph or network embedding have been proposed in recent years. Deep-Walk and Node2Vec generate truncated random walks on graphs and apply the Skip Gram by treating the walks as sentences (Perozzi et al., 2014; Grover & Leskovec, 2016). LINE learns the node embeddings by preserving the first-order and second-order proximities (Tang et al., 2015). Following LINE, SDNE leverages deep neural networks to capture the highly non-linear structure (Wang et al., 2016). Other methods construct a particular objective matrix and use matrix factorization techniques to generate embeddings, e.g., GraRep (Cao et al., 2015) and NetMF (Qiu et al., 2018). This also led to the proliferation of network embedding methods for information-rich graphs, including heterogeneous information networks (Chang et al., 2015; Dong et al., 2017) and attributed graphs (Pan et al., 2016; Liang et al., 2018; Yang et al., 2015; Kipf & Welling, 2017).

On the other hand, there are very few efforts, focusing on the scalability of network embedding (Yang et al., 2017; Huang et al., 2017). First, such efforts are specific to a particular embedding strategy and do not generalize. Second, the scalability of such efforts is limited to moderately sized datasets. Finally, and notably, these efforts at scalability are actually orthogonal to our strategy and can potentially be employed along with our efforts to afford even greater speedup.

The closest work to this paper is the very recently proposed HARP (Chen et al., 2018), which proposes a hierarchical paradigm for graph embedding based on iterative learning methods (e.g., DeepWalk and Node2Vec). However, HARP focuses on improving the quality of embeddings by using the learned embeddings from the previous level as the initialized embeddings for the next level, which introduces a huge computational overhead. Moreover, it is not immediately obvious how a HARP like methodology would be extended to other graph embedding techniques (e.g., GraRep and NetMF) in an agnostic manner since such an approach would necessarily require one to modify the embedding methods to preset their initialized embeddings. In this paper, we focus on designing a general-purpose framework to scale up embedding methods treating them as black boxes.

## 3 Problem Formulation

Let $\mathcal{G} = (V, E)$ be the input graph where $V$ and $E$ are respectively the node set and edge set. Let $A$ be the adjacency matrix of the graph and we assume $\mathcal{G}$ is undirected, though our problem can be easily extended (Chung, 2005; Gleich, 2006; Satuluri & Parthasarathy, 2011) to directed graph. We first define graph embedding:

**Definition 3.1** *Graph Embedding Given a graph $\mathcal{G} = (V, E)$ and a dimensionality d $(d \ll |V|)$, the problem of graph embedding is to learn a d-dimension vector representation for each node in $\mathcal{G}$ so that graph properties are best preserved.*

Following this, a graph embedding method is essentially a mapping function $f : \mathbb{R}^{|V| \times |V|} \mapsto \mathbb{R}^{|V| \times d}$, whose input is the adjacency matrix $A$ (or $\mathcal{G}$) and output is a lower dimension matrix. Motivated by the fact that the majority of graph embedding methods cannot scale

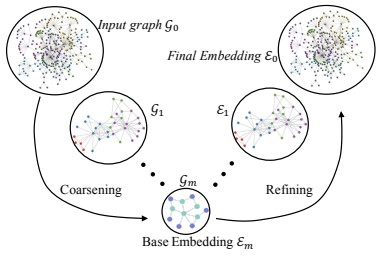
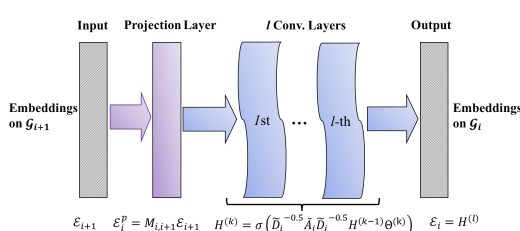

(a) An overview of the multi-level embedding framework.

(b) Architecture of the embeddings refinement model.

Figure 1: MILE framework

to large datasets, we seek to speed up existing graph embedding methods without sacrificing quality. We formulate the problem as:

*Given a graph $\mathcal{G} = (V, E)$ and a graph embedding method $f(\cdot)$, we aim to realize a strengthened graph embedding method $\hat{f}(\cdot)$ so that it is more scalable than $f(\cdot)$ while generating embeddings of comparable or even better quality.*

## 4 METHODOLOGY

MILE framework consists of three key phases: graph coarsening, base embedding, and embeddings refining. Figure 1a shows the overview.

### 4.1 GRAPH COARSENING

In this phase, the input graph $\mathcal{G}$ (or $\mathcal{G}_0$) is repeatedly coarsened into a series of smaller graphs $\mathcal{G}_1, \mathcal{G}_2, ..., \mathcal{G}_m$ such that $|V_0| > |V_1| > ... > |V_m|$. In order to coarsen a graph from $\mathcal{G}_i$ to $\mathcal{G}_{i+1}$, multiple nodes in $\mathcal{G}_i$ are collapsed to form super-nodes in $\mathcal{G}_{i+1}$, and the edges incident on a super-node are the union of the edges on the original nodes in $\mathcal{G}_i$. Here the set of nodes forming a super-node is called a *matching*. We propose a hybrid matching technique containing two matching strategies that can efficiently coarsen the graph while retaining the global structure. An example is shared in Figure 2.

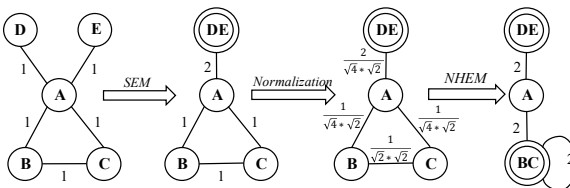
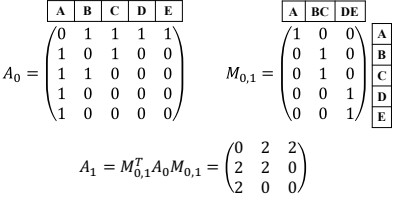

(a) Using SEM and NHEM for graph coarsening

(b) Adjacency matrix and matching matrix

Figure 2: Toy example for illustrating graph coarsening. (a) shows the process of applying Structural Equivalence Matching (SEM) and Normalized Heavy Edge Matching (NHEM) for graph coarsening. (b) presents the adjacency matrix $A_0$ of the input graph, the matching matrix $M_{0,1}$ corresponding to the SEM and NHEM matchings, and the derivation of the adjacency matrix $A_1$ of the coarsened graph using Eq. 2.

**Structural Equivalence Matching (SEM)** : Given two vertices $u$ and $v$ in an unweighted graph $\mathcal{G}$, we call they are *structurally equivalent* if they are incident on the same set of neighborhoods. In figure 2a, node D and E are *structurally equivalent*. The intuition of *matching* structually equivalent nodes is that if two vertices are structurally equivalent, then their node embeddings will be similar.

**Normalized Heavy Edge Matching (NHEM)** : Heavy edge matching is a popular matching method for graph coarsening (Karypis & Kumar, 1998). For an unmatched node $u$ in $\mathcal{G}_i$, its heavy edge matching is a pair of vertices $(u, v)$ such that the weight of the edge between $u$ and $v$ is the largest. In this paper, we propose to normalize the edge weights

when applying heavy edge matching using the formula as follows

$$W_i(u, v) = \frac{A_i(u, v)}{\sqrt{D_i(u, u) \cdot D_i(v, v)}}. \tag{1}$$

Here, the weight of an edge is normalized by the degree of the two vertices on which the edge is incident. Intuitively, it penalizes the weights of edges connected with high-degree nodes. As we will show in Sec. 4.3, this normalization is tightly connected with the graph convolution kernel.

**Hybrid Matching Method** : We use a hybrid of two matching methods above for graph coarsening. To construct $\mathcal{G}_{i+1}$ from $\mathcal{G}_i$, we first find out all the structural equivalence matching (SEM) $\mathcal{M}_1$, where $\mathcal{G}_i$ is treated as an unweighted graph. This is followed by the searching of the normalized heavy edge matching (NHEM) $\mathcal{M}_2$ on $\mathcal{G}_i$. Nodes in each matching are then collapsed into a super-node in $\mathcal{G}_{i+1}$. Note that some nodes might not be matched at all and they will be directly copied to $\mathcal{G}_{i+1}$.

Formally, we build the adjacency matrix $A_{i+1}$ of $\mathcal{G}_{i+1}$ through matrix operations. To this end, we define the *matching matrix* storing the matching information from graph $\mathcal{G}_i$ to $\mathcal{G}_{i+1}$ as a binary matrix $M_{i,i+1} \in \{0, 1\}^{|V_i| \times |V_{i+1}|}$. The $r$-th row and $c$-th column of $M_{i,i+1}$ is set to 1 if node $r$ in $\mathcal{G}_i$ will be collapsed to super-node $c$ in $\mathcal{G}_{i+1}$, and is set to 0 if otherwise. Each column of $M_{i,i+1}$ represents a matching with the 1s representing the nodes in it. Each unmatched vertex appears as an individual column in $M_{i,i+1}$ with merely one entry set to 1. Following this formulation, we construct the adjacency matrix of $\mathcal{G}_{i+1}$ by using

$$A_{i+1} = M_{i,i+1}^T A_i M_{i,i+1}. \tag{2}$$

## 4.2 Base Embedding on Coarsened Graph

The size of the graph reduces drastically after each iteration of coarsening, halving the size of the graph in the best case. We coarsen the graph for $m$ iterations and apply the graph embedding method $f(\cdot)$ on the coarsest graph $\mathcal{G}_m$. Denoting the embeddings on $\mathcal{G}_m$ as $\mathcal{E}_m$, we have $\mathcal{E}_m = f(\mathcal{G}_m)$. Since our framework is agnostic to the adopted graph embedding method, we can use any graph embedding algorithm for base embedding.

## 4.3 Refinement of Embeddings

The final phase of MILE is the embeddings refinement phase. Given a series of coarsened graph $\mathcal{G}_0, \mathcal{G}_1, \mathcal{G}_2, ..., \mathcal{G}_m$, their corresponding matching matrix $M_{0,1}, M_{1,2}, ..., M_{m-1,m}$, and the node embeddings $\mathcal{E}_m$ on $\mathcal{G}_m$, we seek to develop an approach to derive the node embeddings of $\mathcal{G}_0$ from $\mathcal{G}_m$. To this end, we first study an easier subtask: given a graph $\mathcal{G}_i$, its coarsened graph $\mathcal{G}_{i+1}$, the matching matrix $M_{i,i+1}$ and the node embeddings $\mathcal{E}_{i+1}$ on $\mathcal{G}_{i+1}$, how to infer the embeddings $\mathcal{E}_i$ on graph $\mathcal{G}_i$. Once we solved this subtask, we can then iteratively apply the technique on each pair of consecutive graphs from $\mathcal{G}_m$ to $\mathcal{G}_0$ and eventually derive the node embeddings on $\mathcal{G}_0$. In this work, we propose to use a graph-based neural network model to perform embeddings refinement.

**Graph Convolution Network for Refinement Learning :** Since we know the matching information between the two consecutive graphs $\mathcal{G}_i$ and $\mathcal{G}_{i+1}$, we can easily project the node embeddings from the coarse-grained graph $\mathcal{G}_{i+1}$ to the fine-grained graph $\mathcal{G}_i$ using

$$\mathcal{E}_i^p = M_{i,i+1} \mathcal{E}_{i+1} \tag{3}$$

In this case, embedding of a super-node is directly copied to its original node(s). We call $\mathcal{E}_i^p$ the *projected embeddings* from $\mathcal{G}_{i+1}$ to $\mathcal{G}_i$, or simply *projected embeddings* without ambiguity. While this way of simple projection maintains some information of node embeddings, it has obvious limitations that nodes will share the same embeddings if they are matched and collapsed into a super-node during the coarsening phase. This problem will be more serious when the embedding refinement is performed iteratively from $\mathcal{G}_m, ..., \mathcal{G}_0$. To address this issue, we propose to use a *graph convolution network* for embedding refinement. Specifically, we design a graph-based neural network model $\mathcal{E}_i = \mathcal{R}(\mathcal{E}_i^p, A_i)$, which derives the embeddings $\mathcal{E}_i$ on graph $\mathcal{G}_i$ based on the projected embeddings $\mathcal{E}_i^p$ and the graph adjacency matrix $A_i$.

Given graph $G$ with adjacency matrix $A$, we consider the fast approximation of graph convolution from (Kipf & Welling, 2017). The $k$-th layer of this neural network model is

$$H^{(k)}(X, A) = \sigma \left( \tilde{D}^{-\frac{1}{2}} \tilde{A} \tilde{D}^{-\frac{1}{2}} H^{(k-1)}(X, A) \Theta^{(k)} \right) \qquad (4)$$

where $\sigma(\cdot)$ is an activation function, $\Theta^{(k)}$ is a layer-specific trainable weight matrix, and $H^{(0)}(X, A) = X$. In this paper, we define our embedding refinement model as a $l$-layer graph convolution model

$$\mathcal{E}_i = \mathcal{R}\left(\mathcal{E}_i^p, A_i\right) \equiv H^{(l)}\left(\mathcal{E}_i^p, A_i\right). \qquad (5)$$

The architecture of the refinement model is shown in Figure 1b. The intuition behind this refinement model is to integrate the structural information of the current graph $\mathcal{G}_i$ into the projected embedding $\mathcal{E}_i^p$ by repeatedly performing the spectral graph convolution. Each layer of graph convolution network in Eq. 4 can be regarded as one iteration of embedding propagation in the graph following the re-normalized adjacency matrix $\tilde{D}^{-\frac{1}{2}} \tilde{A} \tilde{D}^{-\frac{1}{2}}$. Note that this re-normalized matrix is well aligned with the way we conduct normalized heavy edge matching in Eq. 1. We next discuss how the weight matrix $\Theta^{(k)}$ is learned.

**Intricacies of Refinement Learning :** The learning of the refinement model is essentially learning $\Theta^{(k)}$ for each $k \in [1, l]$ according to Eq. 4. Here we study how to design the learning task and construct the loss function. Since the graph convolution model $H^{(l)}(\cdot)$ aims to predict the embeddings $\mathcal{E}_i$ on graph $\mathcal{G}_i$, we can directly run a base embedding on $\mathcal{G}_i$ to generate the "ground-truth" embeddings and use the difference between these embeddings and the predicted ones as the loss function for training. We propose to learn $\Theta^{(k)}$ on the coarsest graph and **reuse** them across all the levels for refinement. Specifically, we can define the loss function as the mean square error as follows

$$L = \frac{1}{|V_m|} \left\| \mathcal{E}_m - H^{(l)}(M_{m,m+1}\mathcal{E}_{m+1}, A_m) \right\|^2. \qquad (6)$$

We refer to the learning task associated with the above loss function as *double-base* embedding learning. We point out, however, there are two key drawbacks to this method. First of all, the above loss function requires one more level of coarsening to construct $\mathcal{G}_{m+1}$ and an extra base embedding on $\mathcal{G}_{m+1}$. These two steps, especially the latter, introduce non-negligible overheads to the MILE framework, which contradicts our motivation of scaling up graph embedding. More importantly, $\mathcal{E}_m$ might not be a desirable "ground truth" for the refined embeddings. This is because most of the embedding methods are invariant to an orthogonal transformation of the embeddings, i.e., the embeddings can be rotated by an arbitrary orthogonal matrix (Hamilton et al., 2017). In other words, the embedding spaces of graph $\mathcal{G}_m$ and $\mathcal{G}_{m+1}$ can be totally different since the two base embeddings are learned independently. Even if we follow the paradigm in (Chen et al., 2018) and conduct base embedding on $\mathcal{G}_m$ using the simple projected embeddings from $\mathcal{G}_{m+1}$ ($\mathcal{E}_m^p$) as initialization, the embedding space does not naturally generalize and can drift during re-training. One possible solution is to use an alignment procedure to force the embeddings to be aligned between the two graphs (Hamilton et al., 2016). But it could be very expensive.

In this paper, we propose a very simple method to address the above issues. Instead of conducting an additional level of coarsening, we construct a dummy coarsened graph by simply copying $\mathcal{G}_m$, i.e., $M_{m,m+1} = I$ and $\mathcal{G}_{m+1} = \mathcal{G}_m$. By doing this, we not only reduce one iteration of graph coarsening, but also avoid performing base embedding on $\mathcal{G}_{m+1}$ simply because $\mathcal{E}_{m+1} = \mathcal{E}_m$. Moreover, the embeddings of $\mathcal{G}_m$ and $\mathcal{G}_{m+1}$ are guaranteed to be in the same space in this case without any drift. With this strategy, we change the loss function for model learning as follows

$$L = \frac{1}{|V_m|} \left\| \mathcal{E}_m - H^{(l)}(\mathcal{E}_m, A_m) \right\|^2. \qquad (7)$$

With the above loss function, we adopt gradient descent with back-propagation to learn the parameters $\Theta^{(k)}$, $k \in [1, l]$. In the subsequent refinement steps, we apply the same set of parameters $\Theta^{(k)}$ to infer the refined embeddings. We point out that the training of the refinement model is rather efficient as it is done on the coarsest graph. The embeddings

refinement process involves merely sparse matrix multiplications using Eq. 5 and is relatively affordable compared to conducting embedding on the original graph. With these different components, we summarize the whole algorithm of our MILE framework in Algorithm 1. The appendix contains the time complexity of the algorithm in Section A.2

---

**Algorithm 1** Multi-Level Algorithm for Graph Embedding

---

**Input**: A input graph $\mathcal{G}_0 = (V_0, E_0)$, # coarsening levels $m$, and a base embedding method $f(\cdot)$.
**Output**: Graph embeddings $\mathcal{E}_0$ on $\mathcal{G}_0$.

1: Coarsen $\mathcal{G}_0$ into $\mathcal{G}_1, \mathcal{G}_2, ..., \mathcal{G}_m$ using proposed hybrid matching method.
2: Perform base embedding on the coarsest graph $\mathcal{G}_m$ (See Section. 4.2).
3: Learn the weights $\Theta^{(k)}$ using the loss function in Eq. 7.
4: **for** $i = (m-1)...0$ **do**
5:     Compute the projected embeddings $\mathcal{E}_i^p$ on $\mathcal{G}_i$.
6:     Use Eq. 4 and Eq. 5 to compute refined embeddings $\mathcal{E}_i$.
7: Return graph embeddings $\mathcal{E}_0$ on $\mathcal{G}_0$.

---

## 5 Experiments and Analysis

### 5.1 Experimental Configuration

The datasets used in our experiments is shown in Table 1. Yelp dataset is preprocessed by us following similar procedures in (Huang et al., 2017)[1]. To demonstrate that MILE can work with different graph embedding methods , we explore several popular methods for graph embedding, mainly, DeepWalk (Perozzi et al., 2014), Node2vec (Grover & Leskovec, 2016), Line (Tang et al., 2015), GraRep (Cao et al., 2015) and NetMF (Qiu et al., 2018). To evaluate the quality of the embeddings, we follow the typical method in existing work to perform multi-label node classification (Perozzi et al., 2014; Grover & Leskovec, 2016).

| Dataset | # Nodes | # Edges | # Classes |
|---------|---------|---------|-----------|
| PPI | 3,852 | 38,705 | 50 |
| Blog | 10,312 | 333,983 | 39 |
| Flickr | 80,513 | 5,899,882 | 195 |
| YouTube | 1,134,890 | 2,987,624 | 47 |
| Yelp | 8,938,630 | 39,821,123 | 22 |

Table 1: Dataset Information

### 5.2 MILE Framework Performance

We first evaluate the performance of our MILE framework when applied to different graph embedding methods. Figure 3 summarizes the performance of MILE on different datasets with various base embedding methods on various coarsening levels[2] (exact numbers can be seen in Table 3 of Appendix). Note that $m=0$ corresponds to original embedding method. We make the following observations:

- **MILE is scalable**. MILE greatly boosts the speed of the explored embedding methods. With a single level of coarsening ($m=1$), we are able to achieve speedup ranging from $1.5\times$ to $3.4\times$ (on PPI, Blog, and Flickr) while improving qualitative performance. Larger speedups are typically observed on GraRep and NetMF. Increasing the coarsening level $m$ to 2, the speedup increases further (up to $14.4\times$), while the quality of the embeddings is comparable with the original methods reflected by Micro-F1. On YouTube, for the coarsening levels 6 and 8, we observe more than $10\times$ speedup for DeepWalk, Node2Vec and LINE. For NetMF on YouTube, the speedup is even larger – original NetMF runs out of memory within 9.5 hours while MILE (NetMF) only takes around 20 minutes ($m = 8$).

- **MILE improves quality**. For the smaller coarsening levels across all the datasets and methods, MILE-enhanced embeddings almost always offer a qualitative improvement over

---

[1]Raw data: `https://www.yelp.com/dataset_challenge/dataset`
[2]We discuss the results of Yelp later.

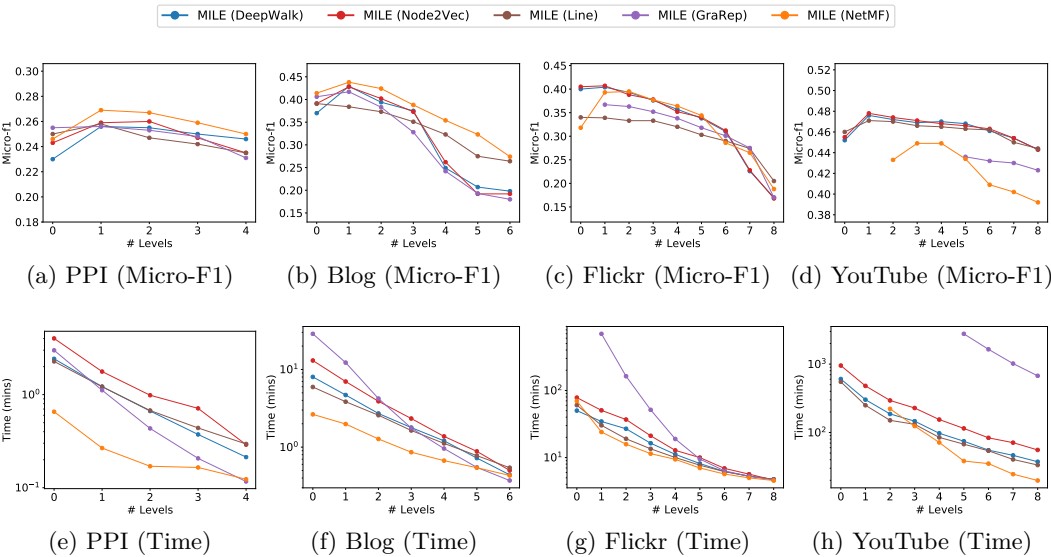

(a) PPI (Micro-F1)  (b) Blog (Micro-F1)  (c) Flickr (Micro-F1)  (d) YouTube (Micro-F1)

(e) PPI (Time)  (f) Blog (Time)  (g) Flickr (Time)  (h) YouTube (Time)

Figure 3: Changes in performance as the number of coarsening levels in MILE increases (best viewed in color). Micro-F1 and running-time are reported in the first and second row respectively. Running time in minutes is shown in logarithm scale. Note that $\#$ level $= 0$ represents the original embedding method without using MILE. Lines/points are missing for algorithms that use over 128 GB of RAM.

the original embedding method as evaluated by the Micro-F1 score (as high as 24.2% while many others also show a 10%+ increase). Examples include MILE (DeepWalk, $m = 1$) on Blog/PPI, MILE (Line, $m = 1$) on PPI and MILE (NetMF, $m = 1$) on PPI/Blog/Flickr. Even with higher number of coarsening level ($m = 2$ for PPI/Blog/Flickr; $m = 6, 8$ for YouTube), MILE in addition to being much faster can still improve, qualitatively, over the original methods on most of the datasets, e.g., MILE(NetMF, $m = 2$) $\gg$ NETMF on PPI, Blog, and Flickr. We conjecture the observed improvement on quality is because the embeddings begin to rely on a more holistic view of the graph.

- **MILE supports multiple embedding strategies.** We make some embedding-specific observations here. We observe that MILE consistently improves both the quality and the efficiency of NetMF on all four datasets (for YouTube, NetMF runs out of memory). For the largest dataset, the speedups afforded exceed 30-fold. We observe that for GraRep, while speedups with MILE are consistently observed, the qualitative improvements, if any, are smaller (for both YouTube and Flickr, the base method runs out of memory). For Line, even though its time complexity is linear to the number of edges (Tang et al., 2015), applying MILE framework on top of it still generates significant speed-up (likely due to the fact that the complexity of Line contains a larger constant factor $k$ than MILE). On the other hand, MILE on top of Line generates better quality of embeddings on PPI and YouTube while falling a bit short on Blog and Flickr. For DeepWalk and Node2Vec, we again observe consistent improvements in scalability (up to 11-fold on the larger datasets) as well as quality using MILE with a few levels of coarsening. However, when the coarsening level is increased, the additional speedup afforded (up to 17-fold) comes at a mixed cost to quality (micro-F1 drops slightly).

- **Impact of varying coarsening levels on MILE.** When coarsening level $m$ is small, MILE tends to significantly improve the quality of embeddings while taking much less time. From $m = 0$ to $m = 1$, we see a clear jump of the Micro-F1 score on all the datasets across the base embedding methods. This observation is more evident on larger datasets (Flickr and YouTube). On YouTube, MILE (DeepWalk) with $m$=1 increases the Micro-F1 score by 5.3% while only consuming half of the time compared to the original DeepWalk. MILE (DeepWalk) continues to generate embeddings of better quality than DeepWalk until $m = 7$, where the speedup is 13×. As the coarsening level $m$ in MILE increases, the running time drops dramatically while the quality of embeddings only decreases slightly.

| | PPI | | Blog | |
|---|---|---|---|---|
| | Mi-F1 | Time | Mi-F1 | Time |
| DeepWalk (DW) | 23.0 | 2.4 | 37.0 | 8.0 |
| MILE (DW) | 25.6 | 1.2 | 42.9 | 4.6 |
| HARP (DW) | 24.1 | 3.0 | 41.3 | 9.8 |
| Node2Vec (NV) | 24.3 | 4.0 | 39.1 | 13.0 |
| MILE (NV) | **25.9** | 1.7 | **42.8** | 6.9 |
| HARP (NV) | 22.3 | 3.9 | 36.2 | 13.16 |
| | Flickr | | YouTube | |
| | Mi-F1 | Time | Mi-F1 | Time |
| DeepWalk | 40.0 | 50.0 | 45.2 | 604.8 |
| MILE (DW) | 40.4 | 34.4 | 46.1 | 55.2 |
| HARP (DW) | 40.6 | 78.2 | 46.6 | 1727.7 |
| Node2Vec | 40.5 | 78.2 | 45.5 | 951.2 |
| MILE (NV) | **40.7** | 50.5 | 46.3 | 83.5 |
| HARP (NV) | 40.5 | 101.1 | **47.2** | 1981.3 |

Table 2: MILE vs. HARP

Figure 4: Running MILE on Yelp dataset.

The running time decreases at an almost exponential rate (logarithm scale on the y-axis in the second row of Figure 3). On the other hand, the Micro-F1 score descends much more slowly (the first row of Figure 3). most of which are still better than the original methods. This shows that MILE can not only consolidates the existing embedding methods, but also provides nice trade-off between effectiveness and efficency.

## 5.3 COMPARING MILE WITH HARP

HARP is a multi-level method primarily for improving the quality of graph embeddings. We compare HARP with our MILE framework using DeepWalk and Node2vec as the base embedding methods[3]. Table 2 shows the performance of these two methods on the four datasets (coarsening level is 1 on PPI/Blog/Flickr and 6 on YouTube). From the table we can observe that MILE generates embeddings of comparable quality with HARP. MILE performs much better than HARP on PPI and Blog, marginally better on Flickr and marginally worse on YouTube. However, MILE is significantly faster than HARP on all the four datasets (e.g. on YouTube, MILE affords a $31\times$ speedup). This is because HARP requires running the whole embedding algorithm on each coarsened graph, which introduces a **huge computational overhead**. Note that for PPI and BLOG – MILE with NetMF (not shown) as its base embeddings produces the best micro-F1 of 26.9 and 43.8, respectively. This shows another advantage of MILE - agnostic to the base embedding when compared with HARP.

## 5.4 MILE: LARGE GRAPH EMBEDDING

We now explore the scalability of MILE on the large Yelp dataset. None of the five graph embedding methods studied in this paper can successfully conduct graph embedding on Yelp within 60 hours on a modern machine with 28 cores and 128 GB RAM. Even extending the run-time deadline to 100 hours, we see DeepWalk and Line barely finish. Leveraging the proposed MILE framework now makes it much easier to perform graph embedding on this scale of datasets (see Figure 4 for the results). We observe that MILE significantly reduces the running time and improves the Micro-F1 score. For example, Micro-F1 score of original DeepWalk and Line are 0.640 and 0.625 respectively, which all take more than 80 hours. But using MILE with $m = 4$, the micro-F1 score improves to 0.643 (DeepWalk) and 0.642 (Line) while achiving speedups of around $1.6\times$. Moreover, MILE reduces the running time of DeepWalk from 53 hours (coarsening level 4) to 2 hours (coarsening level 22) while reducing the Micro-F1 score just by 1% (from 0.643 to 0.634). Meanwhile, there is no change in the Micro-F1 score from coarsening level 4 to 10, where the running time is improved by a factor of two. These results affirm the power of the proposed MILE framework on scaling up graph embedding algorithms while generating quality embeddings.

## 6 CONCLUSION

In this work, we propose a novel multi-level embedding (MILE) framework to scale up graph embedding techniques, without modifying them. Our framework incorporates existing embedding techniques as black boxes, and significantly improves the scalability of extant methods by reducing both the running time and memory consumption. Additionally, MILE also provides a lift in the quality of node embeddings in most of the cases. A fundamental contribution of MILE is its ability to learn a refinement strategy that depends on both the underlying graph properties and the embedding method in use. In the future, we plan to generalize MILE for information-rich graphs and employing MILE for more applications.

---

[3]`https://github.com/GTmac/HARP`

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

# A APPENDIX

## A.1 EXPERIMENTAL CONFIGURATION DETAILS

### A.1.1 DATASETS

The details about the datasets used in our experiments are :

- **PPI** is a Protein-Protein Interaction graph constructed based on the interplay activity between proteins of Homo Sapiens, where the labels represent biological states.
- **Blog** is a network of social relationship of bloggers on BlogCatalog and the labels indicate interests of the bloggers.
- **Flickr** is a social network of the contacts between users on flickr.com with labels denoting the interest groups.
- **YouTube** is a social network between users on YouTube, where labels represent genres of groups subscribed by users.
- **Yelp** is a social network of friends on Yelp and labels indicate the business categories on which the users review.

### A.1.2 BASELINE METHODS

**Baseline Methods:** To demonstrate that MILE can work with different graph embedding methods, we explore several popular methods for graph embedding.

- **DeepWalk** (DW) (Perozzi et al., 2014): Following the original work (Perozzi et al., 2014), we set the length of random walks as 80, number of walks per node as 10, and context windows size as 10.
- **Node2Vec** (NV) (Grover & Leskovec, 2016): We use the same setting as DeepWalk for those common hyper-parameters while setting $p = 4.0$ and $q = 1.0$, which we found empirically to generate better results across all the datasets.
- **Line** (LN) (Tang et al., 2015): This method aims at preserving first-order and second-order proximities and has been applied on large-scale graph. We learn the first-order and second-order embeddings respectively and concatenate them to a unified embedding.
- **GraRep** (GR) (Cao et al., 2015): This method considers different powers (up to $k$) of the adjacency matrix to preserve higher-order graph proximity for graph embedding. It uses SVD decomposition to generate the low-dimensional representation of nodes. We set $k = 4$ as suggested in the original work.
- **NetMF** (NM) (Qiu et al., 2018): It is a recent effort that supports graph embedding via matrix factorization. We set the window size to 10 and the rank $h$ to 1024, and lever the approximate version, as suggested and reported by the authors.

### A.1.3 MILE-SPECIFIC SETTINGS

For all the above base embedding methods, we set the embedding dimensionality $d$ as 128. When applying our MILE framework, we vary the coarsening levels $m$ from 1 to 10 whenever possible. For the graph convolution network model, the self-loop weight $\lambda$ is set to 0.05, the number of hidden layers $l$ is 2, and $\tanh(\cdot)$ is used as the activation function, the learning rate is set to 0.001 and the number of training epochs is 200. The Adam Optimizer is used for model training.

### A.1.4 SYSTEM SPECIFICATION

The experiments were conducted on a machine running Linux with an Intel Xeon E5-2680 CPU (28 cores, 2.40GHz) and 128 GB of RAM. We implement our MILE framework in `Python`. Our code and data are will be available for the replicability purpose. For all the

five base embedding methods, we adapt the original code from the authors[4]. We additionally use TensorFlow package for the embeddings refinement learning component. We lever the available parallelism (on 28 cores) for each method (e.g., the generation of random walks in DeepWalk and Node2Vec, the training of the refinement model in MILE, etc.).

### A.1.5 Evaluation Metrics

To evaluate the quality of the embeddings, we follow the typical method in existing work to perform multi-label node classification (Perozzi et al., 2014; Grover & Leskovec, 2016). Specifically, after the graph embeddings are learned for nodes (label is not used for this part), we run a 10-fold cross validation using the embeddings as features and report the average Micro-F1 and average Macro-F1. We also record the end-to-end wallclock time consumed by each method for scalability comparisons.

### A.2 Time Complexity

It is non-trivial to derive the exact time complexity of MILE as it is dependent on the graph structure, the chosen base embedding method, and the convergence rate of the GCN model training. Here, we provide a rough estimation of the time complexity. For simplicity, we assume the number of vertices and the number of edges are reduced by factor $\alpha$ and $\beta$ respectively at each step of coarsening ($\alpha > 1.0$ and $\beta > 1.0$), i.e., $V_i = \frac{1}{\alpha}V_{i-1}$ and $E_i = \frac{1}{\beta}E_{i-1}$. (we found $\alpha$ and $\beta$ in range $[1.5, 2.0]$, empirically). With $m$ levels of coarsening, the coarsening complexity is approximately $O((1 - 1/\beta^m)/(1 - 1/\beta) \times E))$ and since $1/\beta^m$ is small, the complexity reduces to $O(\frac{\beta}{\beta-1} \times E)$. For the base embedding phase, if the embedding algorithm has time complexity of $T(V, E)$, the complexity of the base embedding phase is $T(\frac{V}{\alpha^m}, \frac{E}{\beta^m})$. For the refinement phase, the time complexity can be divided into two parts, i.e. the GCN model training and the embedding inference applying the GCN model. The former has similar complexity as the original GCN and can be denoted as $O(k_1 * \frac{E}{\beta^m})$ (Kipf & Welling, 2017), where $k_1$ is a small constant related to embedding dimensionality and the number of training epochs. The embedding inference part is simply sparse matrix multiplication using Eq. 4 with time complexity $O(k_2 * E_i)$ when refining the embeddings on graph $\mathcal{G}_i$, where $k_2$ is an even smaller constant ($k_2 < k_1$). As a result, the time complexity of the whole refinement phase is $O(k_1 * \frac{E}{\beta^m} + k_2 * (E + \frac{E}{\beta^1}... + \frac{E}{\beta^{m-1}})) \approx O(k_3 * E)$ where $k_3$ is a small constant.

Overall, for an embedding algorithm of time complexity $T(V, E)$, the MILE framework can reduce it to be $T(\frac{V}{\alpha^m}, \frac{E}{\beta^m}) + O(k * E)$. This is a significant improvement considering $T(V, E)$ is usually very large. The reduction in time complexity is attributed to the fact that we run the embedding learning and refinement model training at the coarsest graph. In addition, the overhead introduced by the coarsening phase and recursive embedding refinement is relatively small (linear to the number of edges $E$). Note that the constant factor $k$ in the complexity term is usually small and we empirically found it to be in the scale of tens. Because of this, even when the complexity of the original embedding algorithm is linear to $E$, our MILE framework could still potentially speed up the embedding process because the complexity of MILE contains a smaller constant factor $k$ (see Sec. 5.2 for the experiment of applying MILE on LINE).

Furthermore, it is worth noting that many of the existing embedding strategies involve hyperparameters tunning for the best performance, especially for those methods based on neural networks (e.g., DeepWalk, Node2Vec, etc.). This in turn requires the algorithm to be run repeatedly – hence any savings in runtime by applying MILE are magnified across multiple runs of the algorithm with different hyper-parameter settings.

---

[4]DeepWalk: `https://github.com/phanein/deepwalk`;
Node2Vec: `http://snap.stanford.edu/node2vec/`;
Line: `https://github.com/tangjianpku/LINE` ;
GraRep: `https://github.com/thunlp/OpenNE`;
NetMF: `https://github.com/xptree/NetMF`

## A.3 MILE Performance

The detailed information about performance evaluation is available in Table 3.

| Method | Micro-F1 | Macro-F1 | Time (mins) |
|---|---|---|---|
| DeepWalk | 23.0 | 18.6 | 2.42 |
| MILE (DeepWalk, $m = 1$) | 25.6(11.3%↑) | 20.4(9.7%↑) | 1.22(2.0×) |
| MILE (DeepWalk, $m = 2$) | 25.5(10.9%↑) | 20.7(11.3%↑) | 0.67(3.6×) |
| Node2Vec | 24.3 | 19.6 | 4.01 |
| MILE (Node2Vec, $m = 1$) | 25.9(6.6%↑) | 20.6(5.1%↑) | 1.77(2.3×) |
| MILE (Node2Vec, $m = 2$) | 26.0(7.0%↑) | 21.1(7.7%↑) | 0.98(4.1×) |
| Line | 25.0 | 19.5 | 2.27 |
| MILE (Line, $m = 1$) | 25.8 (3.2%↑) | 19.8 (1.5%↑) | 1.22 (1.9×) |
| MILE (Line, $m = 2$) | 24.7 (-1.2%↓) | 19.0 (-2.6%↓) | 0.68 (3.3×) |
| GraRep | 25.5 | 20.0 | 2.99 |
| MILE (GraRep, $m = 1$) | 25.6(0.4%↑) | 19.8(-1.0%↓) | 1.11(2.7×) |
| MILE (GraRep, $m = 2$) | 25.3(-0.8%↓) | 19.5(-2.5%↓) | 0.43(6.9×) |
| NetMF | 24.6 | 20.1 | 0.65 |
| MILE (NetMF, $m = 1$) | **26.9**(9.3%↑) | **21.6**(7.5%↑) | 0.27(2.5×) |
| MILE (NetMF, $m = 2$) | 26.7(8.5%↑) | 21.1(5.0%↑) | 0.17(3.9×) |

(a) PPI Dataset

| Method | Micro-F1 | Macro-F1 | Time (mins) |
|---|---|---|---|
| DeepWalk | 37.0 | 21.0 | 8.02 |
| MILE (DeepWalk, $m = 1$) | 42.9(15.9%↑) | 27.0(28.6%↑) | 4.69(1.7×) |
| MILE (DeepWalk, $m = 2$) | 39.4(6.5%↑) | 23.5(11.9%↑) | 2.71(3.0×) |
| Node2Vec | 39.1 | 23.0 | 13.04 |
| MILE (Node2Vec, $m = 1$) | 42.8(9.5%↑) | 26.4(14.8%↑) | 6.99(1.9×) |
| MILE (Node2Vec, $m = 2$) | 40.2(2.8%↑) | 23.9(3.9%↑) | 3.89(3.4×) |
| Line | 39.1 | 22.6 | 5.95 |
| MILE (Line, $m = 1$) | 38.4 (-1.8%↓) | 21.0 (-7.0%↓) | 3.84 (1.55×) |
| MILE (Line, $m = 2$) | 37.3 (-4.6%↓) | 19.6 (-13.2%↓) | 2.58 (2.31×) |
| GraRep | 40.6 | 23.3 | 28.76 |
| MILE (GraRep, $m = 1$) | 41.7(2.7%↑) | 24.0(3.0%↑) | 12.25(2.3×) |
| MILE (GraRep, $m = 2$) | 38.3(-5.7%↓) | 20.4(-12.4%↓) | 4.22(6.8×) |
| NetMF | 41.4 | 25.0 | 2.64 |
| MILE (NetMF, $m = 1$) | **43.8**(5.8%↑) | **27.6**(10.4%↑) | 1.98(1.3×) |
| MILE (NetMF, $m = 2$) | 42.4(2.4%↑) | 25.5(2.0%↑) | 1.27(2.1×) |

(b) Blog Dataset

| Method | Micro-F1 | Macro-F1 | Time (mins) |
|---|---|---|---|
| DeepWalk | 40.0 | 26.5 | 50.08 |
| MILE (DeepWalk, $m = 1$) | 40.4(1.0%↑) | 27.3(3.0%↑) | 34.48(1.5×) |
| MILE (DeepWalk, $m = 2$) | 39.3(-1.8%↓) | 26.1(-1.5%↓) | 26.88(1.9×) |
| Node2Vec | 40.5 | 27.3 | 78.21 |
| MILE (Node2Vec, $m = 1$) | **40.7**(0.5%↑) | **27.7**(1.5%↑) | 50.54(1.5×) |
| MILE (Node2Vec, $m = 2$) | 38.8(-4.2%↓) | 25.8(-5.5%↓) | 36.85(2.1×) |
| Line | 34.0 | 14.5 | 60.42 |
| MILE (Line, $m = 1$) | 33.9 (-0.3%↓) | 13.8 (-4.8%↓) | 30.24 (2.00×) |
| MILE (Line, $m = 2$) | 33.3 (-2.1%↓) | 12.9 (-11.0%↓) | 19.05 (3.17×) |
| GraRep | N/A | N/A | > 2343.37 |
| MILE (GraRep, $m = 1$) | 36.7 | 18.6 | 697.39(>3.4×) |
| MILE (GraRep, $m = 2$) | 36.3 | 18.6 | 163.05(>14.4×) |
| NetMF[5] | 31.8 | 14.0 | 69.72 |
| MILE (NetMF, $m = 1$) | 39.3(23.6%↑) | 24.5(75.0%↑) | 24.03(2.9×) |
| MILE (NetMF, $m = 2$) | 39.5(24.2%↑) | 25.9(85.0%↑) | 15.84(4.4×) |

(c) Flickr Dataset

| Method | Micro-F1 | Macro-F1 | Time (mins) |
|---|---|---|---|
| DeepWalk | 45.2 | 34.7 | 604.83 |
| MILE (DeepWalk, $m = 6$) | 46.1(2.0%↑) | **38.5**(11.0%↑) | 55.20(11.0×) |
| MILE (DeepWalk, $m = 8$) | 44.3(-2.0%↓) | 35.3(1.7%↑) | 37.35(16.2×) |
| Node2Vec | 45.5 | 34.6 | 951.27 |
| MILE (Node2Vec, $m = 6$) | **46.3**(1.8%↑) | 38.3(10.7%↑) | 83.52(11.4×) |
| MILE (Node2Vec, $m = 8$) | 44.3(-2.6%↓) | 35.8(3.5%↑) | 55.55(17.1×) |
| Line | 46.0 | 35.0 | 583.37 |
| MILE (Line, $m = 6$) | 46.2 (0.4%↑) | 36.2 (3.4%↑) | 53.97 (10.81×) |
| MILE (Line, $m = 8$) | 44.4 (-3.5%↓) | 35.7 (2.0%↑) | 33.41 (17.46×) |
| GraRep | N/A | N/A | > 3167.00 |
| MILE (GraRep, $m = 6$) | 43.2 | 32.7 | 1644.89(>1.9×) |
| MILE (GraRep, $m = 8$) | 42.3 | 30.9 | 673.95(>4.7×) |
| NetMF | N/A | N/A | > 574.75 |
| MILE (NetMF, $m = 6$) | 40.9 | 27.8 | 35.22(>16.3×) |
| MILE (NetMF, $m = 8$) | 39.2 | 25.5 | 19.22(>29.9×) |

(d) YouTube Dataset

Table 3: Performance of MILE. DeepWalk, Node2Vec, GraRep, and NetMF denotes the original method without using our MILE framework. $m$ is the number of coarsening levels. The numbers within the parenthesis by the reported Micro-F1 and Macro-F1 scores are the *relative percentage* of change compared to the original method Numbers along with "×" is the speedup compared to the original method. "N/A" indicates the method runs out of memory and we show the amount of running time spent when it happens.

## A.4 MILE Drilldown: Design Choices

We now study the role of the design choices we make within the MILE framework related to the coarsening and refinement procedures described. To this end, we examine alternative design choices and systematically examine their performance. The alternatives we consider are:

- **Random Matching (MILE-rm)**: For each iteration of coarsening, we repeatedly pick a random pair of connected nodes as a match and merge them into a super-node until no more matching can be found. The rest of the algorithm is the same as our MILE.

- **Simple Projection (MILE-proj)**: We replace our embedding refinement model with a simple projection method. In other words, we directly copy the embedding of a super-node to its original node(s) without any refinement (see Eq. 3).

- **Averaging Neighborhoods (MILE-avg)**: For this baseline method, the refined embedding of each node is a weighted average node embeddings of its neighborhoods (weighted by the edge weights). This can be regarded as an embeddings propagation method. We add self-loop to each node[6] and conduct the embeddings propagation for two rounds.

- **Untrained Refinement Model (MILE-untr)**: Instead of training the refinement model to minimize the loss defined in Eq. 7, this baseline merely uses a fixed set of values for parameters $\Theta^{(k)}$ without training (values are randomly generated; other parts of the model in Eq. 4 are the same, including $\tilde{A}$ and $\tilde{D}$).

---

[6]Self-loop weights are tuned to the best performance.

| | PPI | | Blog | | Flickr | | YouTube | |
|---|---|---|---|---|---|---|---|---|
| | Mi-F1 | Time | Mi-F1 | Time | Mi-F1 | Time | Mi-F1 | Time |
| DeepWalk | 23.0 | 2.42 | 37.0 | 8.02 | 40.0 | 50.08 | 45.2 | 604.83 |
| MILE (DW) | **25.6** | 1.22 | **42.9** | 4.69 | **40.4** | 34.48 | **46.1** | 55.20 |
| MILE-rm (DW) | 25.3 | 1.01 | 40.4 | 3.62 | 38.9 | 26.67 | 44.9 | 55.10 |
| MILE-proj (DW) | 20.9 | 1.12 | 34.5 | 3.92 | 35.5 | 25.99 | 40.7 | 53.97 |
| MILE-avg (DW) | 23.5 | 1.07 | 37.7 | 3.86 | 37.2 | 25.99 | 41.4 | 55.26 |
| MILE-untr (DW) | 23.5 | 1.08 | 35.5 | 3.96 | 37.6 | 26.02 | 41.8 | 54.52 |
| MILE-2base (DW) | 25.4 | 2.22 | 35.6 | 6.74 | 37.7 | 53.32 | 41.6 | 94.74 |
| MILE-gs (DW) | 22.4 | 2.03 | 35.3 | 6.44 | 36.4 | 44.81 | 43.6 | 394.72 |
| NetMF | 24.6 | 0.65 | 41.4 | 2.64 | 31.8 | 69.72 | N/A | >574 |
| MILE (NM) | **26.9** | 0.27 | **43.8** | 1.98 | **39.3** | 24.03 | **40.9** | 35.22 |
| MILE-rm (NM) | 25.2 | 0.22 | 41.0 | 1.69 | 37.6 | 20.00 | 39.6 | 33.52 |
| MILE-proj (NM) | 23.5 | 0.12 | 38.7 | 1.06 | 34.5 | 15.10 | 26.4 | 26.48 |
| MILE-avg (NM) | 24.5 | 0.13 | 39.9 | 1.05 | 36.4 | 14.86 | 26.4 | 27.71 |
| MILE-untr (NM) | 24.8 | 0.13 | 39.4 | 1.08 | 36.4 | 15.23 | 30.2 | 27.20 |
| MILE-2base (NM) | 26.6 | 0.29 | 41.3 | 2.33 | 37.7 | 31.65 | 34.7 | 55.18 |
| MILE-gs (NM) | 24.8 | 1.08 | 40.0 | 3.70 | 35.1 | 34.25 | 36.4 | 345.28 |

Table 4: Comparisons of graph embeddings between MILE and its variants. Except for the original methods (DeepWalk and NetMF), the number of coarsening level $m$ is set to 1 on PPI/Blog/Flickr and 6 on YouTube. Mi-F1 is the Micro-F1 score in $10^{-2}$ scale while Time column shows the running time of the method in minutes. "N/A" denotes the method consumes more than 128 GB RAM.

- **Double-base Embedding for Refinement Training (MILE-2base)**: This method replaces the loss function in Eq. 7 with the alternative one in Eq. 6 for model training. It conducts one more layer of coarsening and base embedding (level $m + 1$), from which the embeddings are projected to level $m$ and used as the input for model training.

- **GraphSAGE as Refinement Model (MILE-gs)**: It replaces the graph convolution network in our refinement method with GraphSAGE (Hamilton et al., 2017)[7]. We choose max-pooling for aggregation and set the number of sampled neighbors as 100, as suggested by the authors. Also, concatenation is conducted instead of replacement during the process of propagation.

Table 4 shows the comparison of performance on these methods across the four datasets. Here, we focus on using DeepWalk and NetMF for base embedding with a smaller coarsening level ($m = 1$ for PPI, Blog, and Flickr; $m = 6$ for YouTube). Results are similar for the other embedding options we consider. We hereby summarize the key information derived from Table 4 as follows:

- **The matching methods used within MILE offer a qualitative benefit at a minimal cost to execution time.** Comparing MILE with MILE-rm for all the datasets, we can see that MILE generates better embeddings than MILE-rm using either DeepWalk or NetMF as the base embedding method. Though MILE-rm is slightly faster than MILE due to its random matching, its Micro-F1 score and Macro-F1 score are consistently lower than of MILE.

- **The graph convolution based refinement learning methodology in MILE is particularly effective.** Simple projection-based MILE-proj, performs significantly worse than MILE. The other two variants (MILE-avg and MILE-untr) which do not train the refinement model at all, also perform much worse than the proposed method. Note MILE-untr is the same as MILE except it uses a default set of parameters instead of learning those parameters. Clearly, the model learning part of our refinement method is a fundamental contributing factor to the effectiveness of MILE. Through training, the refinement model is tailored to the specific graph under the base embedding method in use. The overhead cost of this learning (comparing MILE with MILE-untr), can vary depending on the base embedding employed (for instance on the YouTube dataset, it is

---

[7]Adapt code from `https://github.com/williamleif/GraphSAGE`

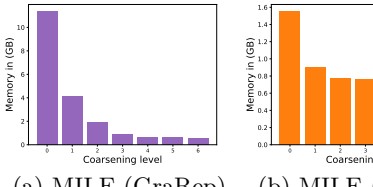

(a) MILE (GraRep)     (b) MILE (NetMF)

Figure 5: Memory consumption of MILE (GraRep) and MILE (NetMF) on Blog with varied coarsening levels.

an insignificant 1.2% on DeepWalk - while being up to 20% on NetMF) but is still worth it due to qualitative benefits (Micro-F1 up from 30.2 to 40.9 with NetMF on YouTube).

- **Graph convolution refinement learning outperforms GraphSAGE.** Replacing the graph convolution network with GraphSAGE for embeddings refinement, MILE-`gs` does not perform as well as MILE. It is also computationally more expensive, partially due to its reliance on embeddings concatenation, instead of replacement, during the process the embeddings propagation (higher model complexity).

- **Double-base embedding learning is not effective.** In Sec. 4.3, we discuss the issues with unaligned embeddings of the double-base embedding method for the refinement model learning. The performance gap between MILE and MILE-`2base` in Table 4 provides empirical evidence supporting our argument. This gap is likely caused by the fact that the base embeddings of level $m$ and level $m + 1$ might not lie in the same embedding space (rotated by some orthogonal matrix) (Hamilton et al., 2017). As a result, using the projected embeddings $\mathcal{E}_m^p$ as input for model training (MILE-`2base`) is not as good as directly using $\mathcal{E}_m$ (MILE). Moreover, Table 4 shows that the additional round of base embedding in MILE-`2base` introduces a non-trivial overhead. On YouTube, the running time of MILE-`2base` is 1.6 times as much as MILE.

## A.5 MILE DRILLDOWN: MEMORY CONSUMPTION

We also study the impact of MILE on reducing memory consumption. For this purpose, we focus on MILE (GraRep) and MILE (NetMF), with GraRep and NetMF as base embedding methods respectively. Both of these are embedding methods based on matrix factorization, which possibly involves a dense objective matrix and could be rather memory expensive. We do not explore DeepWalk and Node2Vec here since their embedding learning methods generate truncated random walks (training data) on the fly with almost negligible memory consumption (compared to the space storing the graph and the embeddings). Figure 5 shows the memory consumption of MILE (GraRep) and MILE(NetMF) as the coarsening level increases on Blog (results on other datasets are similar). We observe that MILE significantly reduces the memory consumption as the coarsening level increases. Even with one level of coarsening, the memory consumption of GraRep and NetMF reduces by 64% and 42% respectively. The dramatic reduction continues as the coarsening level increases until it reaches 4, where the memory consumption is mainly contributed by the storage of the graph and the embeddings. This memory reduction is consistent with our intuition, since both # rows and # columns in the objective matrix reduce almost by half with one level of coarsening.

## A.6 MILE DRILLDOWN: DISCUSSION ON REUSING $\Theta^{(k)}$ ACROSS ALL LEVELS

Similar to GCN, $\Theta^{(k)}$ is a matrix of filter parameters and is of size $d * d$ (where $d$ is the embedding dimensionality). Eq. 4 in this paper defines how the embeddings are propagated during embedding refinements, parameterized by $\Theta^{(k)}$. Intuitively, $\Theta^{(k)}$ defines how different embedding dimensions interact with each other during the embedding propagation. This interaction is dependent on graph structure and base embedding method, which can be learned from the coarsest level. Ideally, we would like to learn this parameter $\Theta^{(k)}$ on every two consecutive levels. But this is not practical since this could be expensive as the graph get more fine-grained (and defeat our purpose of scaling up graph embedding). This trick of "sharing" parameters across different levels is the trade-off between efficiency and

effectiveness. To some extent, it is similar to the original GCN (Kipf & Welling, 2017), where the authors share the same filter parameters $\Theta^{(k)}$ over the whole graph (as opposed to using different $\Theta^{(k)}$ for different nodes; see Eq (6) and (7) in(Kipf & Welling, 2017)). Moreover, we empirically found this works good enough and much more efficient. Table 4 shows that if we do not share $\Theta^{(k)}$ values and use random values for $\Theta^{(k)}$ during refinements, the quality of embedding is much worse (see baseline MILE-untr).

### A.7 MILE Drilldown: Discussion on choice of embedding methods

We wish to point out that we chose the base embedding methods as they are either recently proposed NetMF (introduced in 2018) or are widely used (DeepWalk, Node2Vec, LINE). By showing the performance gain of using MILE on top of these methods, we want to ensure the contribution of this work is of broad interest to the community. We also want to reiterate that these methods are quite different in nature:

- DeepWalk (DW) and Node2vec (N2V) rely on the use of random walks for latent representation of features.
- LINE learns an embedding that directly optimizes a carefully constructed objective function that preserves both first/second order proximity among nodes in the embedding space.
- GraRep constructs multiple objective matrices based on high orders of random walk laplacians, factories each objective matrix to generate embeddings and then concatenates the generated embeddings to form final embedding.
- NetMF constructs an objective matrix based on random walk Laplacian and factorizes the objective matrix in order to generate the embeddings.

Indeed NetMF (Qiu et al., 2018; Levy & Goldberg, 2014) with an appropriately constructed objective matrix has been shown to *approximate* DW, N2V and LINE allowing such be conducting implicit matrix factorization of *approximated* matrices. There are limitations to such approximations (shown in a related context by (Arora et al., 2016)) - the most important one is the requirement of a sufficiently large embedding dimensionality. Additionally, we note that while unification is possible under such a scenario, the methods based on matrix factorization are quite different from the original methods and do place a much larger premium on space (memory consumption) - in fact this is observed by the fact we are unable to run NetMF and GraRep in many cases without incorporating them within MILE.

### A.8 MILE Drilldown: Discussion on extending MILE to directed graphs

Note that as pointed out by (Chung, 2005), one can construct random-walk Laplacians for a directed graph thus incorporating approaches like NetMF to accommodate such solutions. Another simple solution is to symmetrize the graph while accounting for directionality. Once the graph is symmetrized, any of the embedding strategies we discuss can be employed within the MILE framework (including the coarsening technique). There are many ideas for symmetrization of directed graphs (see for example work described by (Gleich, 2006) or (Satuluri & Parthasarathy, 2011).

### A.9 MILE Drilldown: Discussion on effectiveness of SEM

The effectiveness of structurally equivalent matching (SEM) is highly dependent on graph structure but in general 5% - 20% of nodes are structurally equivalent (most of which are low-degree nodes). For example, during the first level of coarsening, YouTube has 172,906 nodes (or 86,453 pairs) out of 1,134,890 nodes that are found to be SEM (around 15%); Yelp has 875,236 nodes (or 437,618 pairs) out of 8,938,630 nodes are SEM (around 10%). In fact, more nodes are involved in SEM as SEM is run iteratively at each coarsening level.

