# OpenReview forum: "MILE: A Multi-Level Framework for Scalable Graph Embedding"
_ICLR.cc/2019/Conference_

### Official Review · AnonReviewer1 · 2018-10-27
**Interesting idea and result**

**Rating:** 6
**Confidence:** 5

**Review:**

This paper proposed a multi-Level framework for learning node embeddings for large-scale graphs. The author first coarsens the graphs into different levels of subgraphs. The low-level subgraphs are obtained with the node embeddings of the higher-level graphs with a graph convolutional neural network. By iteratively applying this procedure, the node embeddings of the original graphs can be obtained. Experimental results on several networks (including one network with ~10M node) prove the effective and efficiency of the proposed method over existing state-of-the-art approaches.

Strength:
- scaling up node embedding methods is a very important and practical problem
- experiments show that the proposed methods seems to be very effective.
Weakness:
- the proposed method seems to be very heuristic
- some claims in the papers are wrong according to existing literatures

Overall, the paper is well written and easy to follow. The proposed method is simple but heuristic.  However, the performance seems to be quite effective according to the experiments. The reasons that why the method works need to be better explained, which can significantly the quality of the paper and its impact in the future.

Details:
-- In the introduction part, "However, such methods rarely scale to large datasets (e.g., graphs with over 1 million nodes) since they are computationally expensive and often memory intensive". This is not TRUE! In the paper of LINE (Tang et al. 2015). It shows the LINE model can easily scale up to networks with one million nodes with a few hours.
-- The authors use Equation (7) to learn the parameters of the graph convolutional neural network. I am really surprised that this method works. Especially the learned parameters are shared across different layers.
-- Have you tried and compared different approaches of graph coarsening?
-- In Figure 2. (a), according to Equation (1), in the second step, the weight of the edge between A and DE should be 2/sqrt(3)*sqrt(4)?

---

> ### Author Response · Authors · 2018-11-14
> **Acknowledging the reviews and our responses**
>
> We thank the reviewer for the review. Please see our responses in detail below.
>
> 1) “some claims in the papers are wrong according to existing literatures”
>
> **Response**:
> We assume the reviewer is referring to the LINE comparison, please see the detailed response below.
>
> ----------------------------------------------------
> 2) “The reasons that why the method works need to be better explained, which can significantly (improve) the quality of the paper and its impact in the future.”
>
> **Response**:
> We conduct a detailed drilldown study which due to lack of space is reported in the Appendix (see Table 5). This drilldown study offers some empirical reasons why we picked the design choices we used which match the intuition described in the main paper.
>
> ----------------------------------------------------
> 3) "However, such methods rarely scale to large datasets (e.g., graphs with over 1 million nodes) since they are computationally expensive and often memory intensive". This is not TRUE! In the paper of LINE (Tang et al. 2015). It shows the LINE model can easily scale up to networks with one million nodes with a few hours.
>
> **Response**:
> We use the word “rarely” in the quote above. We feel this statement is still true (outside of LINE and a couple of other papers very few papers scales to large datasets). Our effort can scale both methods like LINE as well as methods that do not scale particularly well.
>
> A few minor notes -- The paper by Tang et al reported results on a 1TB RAM machine -- we used a 128GB RAM machine.  We also report results on a much larger dataset Yelp.  For all the results, we report the wallclock time of the entire execution.
>
> Finally,  if the reviewer has a specific suggestion on how to rephrase the above statement we are happy to accommodate.
>
> ----------------------------------------------------
> 4) "The authors use Equation (7) to learn the parameters of the graph convolutional neural network. I am really surprised that this method works. Especially the learned parameters are shared across different layers. "
>
> **Response**:
>  Similar to GCN, \Theta is a matrix of filter parameters and is of size dxd (where d is the embedding dimensionality). Eq. (4) in this paper defines how the embeddings are propagated during embedding refinements, parameterized by \Theta. Intuitively,  \Theta defines how different embedding dimensions interact with each other during the embedding propagation. This interaction is dependent on graph structure and base embedding method, which can be learned from the coarsest level.
>
> Ideally, we would like to learn this parameter \Theta on every two consecutive levels. But this is not practical since this could be expensive as the graph get more fine-grained (and defeat our purpose of scaling up graph embedding). This trick of “sharing” parameters across different levels is the trade-off between efficiency and effectiveness. To some extent, it is similar to the original GCN [1], where the authors share the same filter parameters \Theta over the whole graph (as opposed to using different \Theta for different nodes; see Eq (6) and (7) in [1]).  -- We did not include these details due to the limit of space but would be happy to add them in the final version.
>
> Moreover, we empirically found this works good enough and much more efficient. Table 5 shows that if we do not share \Theta values and use random values for \Theta during refinements, the quality of embedding is much worse (see baseline MILE-untr).  We thank the reviewer for this question and we will better explain this in the revised version of the article.
>
> [1] Kipf, Thomas N., and Max Welling. "Semi-supervised classification with graph convolutional networks." ICLR (2017).
>
> ----------------------------------------------------
> 5) "Have you tried and compared different approaches of graph coarsening?"
>
> **Response**:
> Yes -- we did try several ideas -- we included some of these results in Table 5.
>
> ----------------------------------------------------
> 6). "In Figure 2. (a), according to Equation (1), in the second step, the weight of the edge between A and DE should be 2/sqrt(3)*sqrt(4)?"
>
> **Response**:
> Thanks for catching this typo -- it should be in fact 2/(sqrt(4)*sqrt(2)).
> Reasoning:
> The degree of node A is D(A) = 4.
> The degree of node DE is D(DE) = 2.
> So this should be 2/(sqrt(4)*sqrt(2)).

---

### Official Review · AnonReviewer3 · 2018-11-01
**Practically useful, but experiments are not convincing**

**Rating:** 4
**Confidence:** 4

**Review:**

In this submission, the authors propose a three-stage framework for large-scale graph embedding. The proposed method first constructs a small graph by graph coarsening, then applies any existing graph embedding method, and last refines the learned embeddings. It is useful, however, the experimental results are not convincing and cannot support the authors' claims about the proposed method.

First, in many places, the authors claim that the embedding quality of the proposed method is improved. For example, the last sentence of Section 1, and "MILE improves quality" paragraph on Page 7. However, the experimental results fail to support this. As the proposed method is for the large-scale graph, let's focus on the results of YouTube dataset and Yelp dataset first. For Youtube dataset ((d) of Table 2), when m is set to be 8, for all the cases, the performance drops. For Yelp dataset (Figure 3), the authors do not provide Micro-f1 for the original graph (m = 0) or m = 1, 2, so it is hard or impossible to demonstrate that the quality of the proposed method is still good.

Second, the comparison with existing methods is not sufficient. For the most important Yelp dataset (as this dataset fits the motivation scenario (large-scale graph) of this submission), the authors fail to report any comparison. Thus it might not be weak to demonstrate the benefit of the proposed method.

Third, some experiment details are missing. For example, how the authors compute the running time of the proposed method? All the three stages are included? How the authors implement the existing methods? Are these implementations good enough to ensure a fair comparison?

*******
Some other questions:
a) On page 2, the authors mention that the proposed method "can be easily extended to directed graph". However, based on my understanding, directly graph will affect both the graph coarsening and embedding refining steps, and it seems not so easy to extend. Do the authors have the solution and experiments for directed graph? It would be interesting to see such results, which enlarges the application scope of the proposed method.

b) The toy example on page 3 is very clear. However, for real-world graphs, does the proposed graph coarsening work well? For example, one property the proposed method utilizes is "structurally equivalent". What is the percentage of the nodes that can have such property for real-world graphs?

********
Some other comments:
Generally speaking, this submission studies a very practical task. Although the authors claim that the proposed method has great efficiency while the embedding quality is comparable good or even better than the existing methods, I think that there is an efficiency-quality trade-off based on the experimental results in this submission. When m increases, the graph coarsening step causes more information loss, and thus the quality may decrease. Embedding refining step can be regarded as a procedure to reduce such information loss, but may not improve the embedding quality better than the original graph. So to me, it would be more meaningful to study such efficiency-quality trade-off for large-scale graph embedding.

---

> ### Author Response · Authors · 2018-11-14
> **Thank the Reviewer and Our Responses (Part-2)**
>
> 5) "On page 2, the authors mention that the proposed method "can be easily extended to directed graph". However, based on my understanding, directly graph will affect both the graph coarsening and embedding refining steps, and it seems not so easy to extend. Do the authors have the solution and experiments for directed graph? It would be interesting to see such results, which enlarges the application scope of the proposed method."
>
> **Response**:
> Note that as pointed out by Chung et al. [1] one can construct random-walk Laplacians for a directed graph thus incorporating approaches like NetMF to accommodate such solutions.  Another simple solution is to symmetrize the graph while accounting for directionality. Once the graph is symmetrized, any of the embedding strategies we discuss can be employed within the MILE framework (including the coarsening technique). There are many ideas for symmetrization of directed graphs (see for example work described by Gleich in 2006 [2] or Satuluri and Parthasarathy in 2011 [3]).
>
> [1] Chung, Fan. "Laplacians and the Cheeger inequality for directed graphs." Annals of Combinatorics 9, no. 1 (2005): 1-19.
> [2] David Gleich, Hierarchical directed spectral graph partitioning, Information Networks 2006.
> [3] Venu Satuluri and Srinivasan Parthasarathy, Symmetrizations for clustering directed graphs, EDBT 2011.
>
> ----------------------------------------------------
> 6) "The toy example on page 3 is very clear. However, for real-world graphs, does the proposed graph coarsening work well? For example, one property the proposed method utilizes is "structurally equivalent". What is the percentage of the nodes that can have such property for real-world graphs?"
>
> **Response**:
> We included results against strawman coarsening strategies in Table 5 of MILE Drilldown in the Appendix-- see the performance of MILE vs MILE-rm. With regards to how often the structurally equivalent matching (SEM) is effective, this is highly dependent on graph structure but in general 5% ~ 20% of nodes are structurally equivalent (most of which are low-degree nodes). For example, during the first level of coarsening, YouTube has 172,906 nodes (or 86,453 pairs) out of 1,134,890 nodes that are found to be SEM (so ~15%); Yelp has 875,236 nodes (or 437,618 pairs) out of 8,938,630 nodes are SEM (so ~10%). In fact, more nodes are involved in SEM as SEM is run iteratively at each coarsening level.
>
> ----------------------------------------------------
> 7) "Although the authors claim that the proposed method has great efficiency while the embedding quality is comparable good or even better than the existing methods, I think that there is an efficiency-quality trade-off based on the experimental results in this submission. "
>
> **Response**:
> We have addressed this comment above. Again, we kindly remind the reviewer on our results in Figure 4 and the analysis around it in the Appendix. To reiterate, we always see an improvement in quality using MILE for smaller values of m as compared to running the embedding method on the original graph (e.g. small m vs. m=0). After a point, there is an efficiency-quality tradeoff as m increases. This is clearly shown in Figure 4 by comparing m=1 (w/ MILE) vs. m=0 (w/o MILE).
>
> For Yelp at m=0 the micro-F1 is 0.625 and it takes over 80 hours to complete (LINE). For m=22 we are obtaining a micro-F1 of 0.635 and it takes about 2.6 hours to complete. So this is a speedup of over 30 and an improvement of the micro-F1 score. On the other hand, at m=8 (which is using MILE) the speedup is about 2.5 (lower) but with an even better micro-F1 score of 0.642 (similar story on DeepWalk) -- showing nice trade-off property when using MILE but are all much better than the one without using MILE.

---

> ### Author Response · Authors · 2018-11-14
> **Thank the Reviewer and Our Responses (Part-1)**
>
> We thank the reviewer for providing detailed comments. I tried our best to answer the questions below.
> ----------------------------------------------------
> 1) “First, in many places, the authors claim that the embedding quality of the proposed method is improved. For example, the last sentence of Section 1, and "MILE improves quality" paragraph on Page 7. However, the experimental results fail to support this. As the proposed method is for the large-scale graph, let's focus on the results of YouTube dataset and Yelp dataset first. For Youtube dataset ((d) of Table 2), when m is set to be 8, for all the cases, the performance drops. For Yelp dataset (Figure 3), the authors do not provide Micro-f1 for the original graph (m = 0) or m = 1, 2, so it is hard or impossible to demonstrate that the quality of the proposed method is still good. ”
>
> **Response**:
> We do observe MILE improves quality on both YouTube and Yelp.
>
> Regarding YouTube results in Table 2(d), with m=6, we can see some nice quality gain and huge speedups (on DeepWalk, Node2Vec, and LINE) -- comparing to m=0 (i.e., w/o MILE). Of course, as m increases, the quality could drop. What we want to show here is that MILE could push even further on the speedup side with little loss of quality. But if the quality is the first consideration, we would suggest using a smaller coarsening level (e.g. m=6) where both quality and efficiency gain can be achieved. Please note again that Figure 4 in the Appendix reports results for varying values of m across all methods on all datasets included in Table 2. Note that some results of original GraRep and NetMF methods are missing. This is because these methods are memory-intense and run out of memory on our machine (128GB RAM).
>
> Regarding Yelp results in Figure 3, we did not report the performance of original embedding methods (m=0) since these methods either take a substantial amount of time or requires too much memory. However, we just recently finished running LINE and DeepWalk (m=0, i.e. w/o MILE) on Yelp. Our results show Micro-F1 on Yelp with no coarsening (m=0)  of is 0.625 for LINE and 0.640 for DW, and they all take more than 80 hours.  At m=4 the micro-F1 improves to 0.642 (LINE) and 0.643 (DW) -- it stays relatively constant at this micro-F1 till about m=8. From m=8 to m=22 they dip slightly below 0.64. Note that even at m=10 they outperform the quality we achieve at m=0 quite significantly (0.639 vs. 0.625 for LINE; 0.643 vs 0.640 for DW). The above result is consistent with the results on other datasets, where for smaller values of m, quality improves but after a point there is a tradeoff between quality and speed (Figure 4 in Appendix in original submission makes this point).  We will include these results in a revised version of the paper.
>
>
> ----------------------------------------------------
> 2) "Second, the comparison with existing methods is not sufficient."
>
> **Response**:
> We compare across 5 methods and across 5 datasets over a range of settings both in the main paper and in the Appendix. Please also note that in the drilldown experiment in the Appendix we also defend various design choices.
>
> ----------------------------------------------------
> 3) "For the most important Yelp dataset (as this dataset fits the motivation scenario (large-scale graph) of this submission), the authors fail to report any comparison. Thus it might not be weak to demonstrate the benefit of the proposed method."
>
> **Response**:
> We want to kindly remind the reviewer that we report both Micro-F1 comparison and runtime comparison on all five methods evaluated within the MILE framework (see both parts of Fig 3).  We also plan to add a few more updated results of the original LINE and DeepWalk (m=0) as mentioned above.
>
> ----------------------------------------------------
> 4) "Third, some experiment details are missing. For example, how the authors compute the running time of the proposed method? All the three stages are included? How the authors implement the existing methods? Are these implementations good enough to ensure a fair comparison? "
>
> **Response**:
> All of these questions are addressed in the paper but we repeat here for the reviewer’s benefit. We always compare end-to-end wallclock time of all methods (so for all the MILE variants it includes the computation time of all three stages, discussed in Appendix A.1.5). Existing methods are publicly available implementations from the authors’ GitHub repository when available (pointed out in Appendix A.1.4). Keep in mind in each case we are comparing each method with itself, i.e. with and without MILE (at various coarsening levels). MILE is able to scale all of them individually while in many cases also improving quality. Again, please see Figure 4 in the Appendix as well as the results shared above.

---

### Official Review · AnonReviewer2 · 2018-11-03
**Overall Interesting work: clear motivation and nice performance gain**

**Rating:** 7
**Confidence:** 3

**Review:**

This paper proposes a multi-level embedding (MILE) framework, which can be applied on top of existing network embedding methods and helps them scale to large scale networks with faster speed. To get the backbone structure of graph, MILE repeatedly coarsens the graph into smaller ones using a hybrid matching technique, and GCN is used for the refinement of embeddings.

[+] The paper is well-written and the idea is clearly presented.
[+] MILE is able to reduce computational cost while achieving comparable, or sometimes even better embedding quality.
[+] MILE is general enough to apply to different underlying embedding strategies.
[-] Most of the baseline methods are of similar type, since LINE, DeepWalk, node2vec and NetMF can all be unified to matrix factorization framework. There have been many new network embedding methods proposed in the past two years. It would be interesting to see how much MILE can help scale these methods.

Overall, though there have already been hundreds of papers on network embedding in the past 2~3 years, I think this paper can be an interesting addition to this fast-growing area. Therefore, I would recommend to accept it.

---

> ### Author Response · Authors · 2018-11-14
> **Appreciate the comments and we added some clarifications**
>
> We thank the reviewer for the insightful comments. We wish to point out that we chose the base embedding methods as they are either recently proposed (NetMF introduced in 2018, and GraRep) or are widely used (DeepWalk, Node2Vec, LINE etc.). By showing the performance gain of using MILE on top of these methods, we want to ensure the contribution of this work is of broad interest to the community.
>
> We also want to reiterate that these methods are quite different in nature:
> * DeepWalk (DW) and Node2vec (N2V) rely on the use of random walks for latent representation of features.
> * LINE learns an embedding that directly optimizes a carefully constructed objective function that preserves both first/second order proximity among nodes in the embedding space.
> * GraRep constructs multiple objective matrices based on high orders of random walk laplacians, factories each objective matrix to generate embeddings and then concatenates the generated embeddings to form final embedding.
> * NetMF constructs an objective matrix based on random walk Laplacian and factorizes the objective matrix in order to generate the embeddings.
>
> Indeed as the reviewer notes, under a few assumptions [1,2], NetMF with an appropriately constructed objective matrix has been shown to “approximate” DW, N2V and LINE allowing such be conducting implicit matrix factorization of **approximated** matrices. There are limitations to such approximations (shown in a related context by Arora et al [3]) - the most important one is the requirement of a sufficiently large embedding dimensionality.  Additionally, we note that while unification is possible under such a scenario, the methods based on matrix factorization are quite different from the original methods and do place a much larger premium on space (memory consumption) - in fact this is observed by the fact we are unable to run NetMF and GraRep in many cases without incorporating them within MILE (as noted in the paper) and also in one of the other responses below.
>
> In this paper, the base embedding methods are implemented using the original embedding learning algorithm (e.g. DW, N2V, Line) -- which directly are from the authors’ code.
>
> That being said, we really appreciate reviewer’s suggestion of exploring MILE on other types of network embedding. As part of the future work, we will look into how MILE can be used in the case of attributed network embedding. In another response below we also discuss how it can be incorporated in a directed graph setting.
>
> [1] Qiu, Jiezhong, Yuxiao Dong, Hao Ma, Jian Li, Kuansan Wang, and Jie Tang. "Network embedding as matrix factorization: Unifying deepwalk, line, pte, and node2vec." In Proceedings of the Eleventh ACM International Conference on Web Search and Data Mining, pp. 459-467. ACM, 2018.
> [2] Levy, Omer, and Yoav Goldberg. "Neural word embedding as implicit matrix factorization." In Advances in neural information processing systems, pp. 2177-2185. 2014.
> [3] Arora, Sanjeev, Yuanzhi Li, Yingyu Liang, Tengyu Ma, and Andrej Risteski. "A latent variable model approach to pmi-based word embeddings." Transactions of the Association for Computational Linguistics 4 (2016): 385-399.

---

### Author Response · Authors · 2018-11-26
**Revision Notes**

We thank the reviewers for providing insightful reviews. We present below the main changes to the document. Every change is a response to the detailed reviews we received. Specifically we:
* Replaced Table 2 (results on selected coarsening levels, previously in the main body) with Figure 3 (results with all varying coarsening levels; previously in Appendix) in response to multiple reviewer questions. The table is now in Appendix and Figure 3 is now in the main body. Blended “Impact of varying coarsening levels on MILE” (previously in the Appendix A.5) in the main body (Sec 5.2).
* Fixed typo error in Figure 2 as pointed out by the reviewer.
* Updated Figure 4 (added results for m=0 and m=2) and Sec 5.4 in response to reviewer comments and edited associated description.
* Added related literature on how to extend our ideas to the directed graph (Sec. 3 and Appendix A.9) in response to reviewer comments.
* Added  "Discussion on reusing \theta" in A.7 in Appendix in response to reviewer comments.
* Added  "Discussion on the choice of embedding methods"  in A.8 in Appendix in response to reviewer comments.
* Added "Discussion on extending MILE to directed graphs"  in A.9 in Appendix in response to reviewer comments.
* Added "Discussion on the effectiveness of SEM"  in A.10 in Appendix in response to reviewer comments.
* Fixed minor grammatical errors.

---

### Meta-Review · Area_Chair1 · 2018-12-14
**Reject**

**Confidence:** 4
**Recommendation:** Reject

**Metareview:**

Significant spread of scores across the reviewers and unfortunately not much discussion despite prompts from the area chair and the authors. The most positive reviewer is the least confident one. Very close to the decision boundary but after careful consideration by the senior PCs just below the acceptance threshold. There is significant literature already on this topic. The "thought delta" created by this paper and the empirical results are also not sufficient for acceptance.